# Shoulder specific exercise therapy is effective in reducing chronic shoulder pain: A network meta-analysis

**Anelise Silveira**[1]* , **Camila Lima**[1], **Lauren Beaupre**[2,3], **Judy Chepeha**[3],
**Allyson Jones**[2]

**1** University of Alberta, School of Public Health, Edmonton, Alberta, Canada, **2** University of Alberta, Faculty of Rehabilitation Sciences, Edmonton, Alberta, Canada, **3** University of Alberta, Collaborative Orthopaedic Research, Edmonton, Alberta, Canada

☯ These authors contributed equally to this work.
* asilveir@ualberta.ca

## Abstract

### Background

Exercise therapy (ET) is frequently an early treatment of choice when managing shoulder pain, yet evidence on its efficacy to expedite recovery is inconsistent. Moreover, the value of adding adjunct therapies (i.e. injections, manual therapy, electrotherapy) to ET is currently unclear. This study combined both direct and indirect evidence across studies on the effectiveness of ET with/without adjunct therapies compared to usual medical care for adults with chronic shoulder pain.

### Methods and findings

Using a network meta-analysis, randomized control trials comparing ET along with adjunct therapies were identified in MEDLINE, Embase, CINAHL, Sportdiscus, CENTRAL, Conference Proceedings Citation Index-Science, clinicaltrials.gov, and association websites. Outcomes included pain, range of motion (ROM), and health-related quality of life (HRQL) measures in adult patients with chronic shoulder pain. Data analysis used a Frequentist hierarchical model. CINeMA tool assessed the confidence in the results and Cochrane Risk of Bias tool assessed quality of studies.

54 studies primarily from Europe (40.38%) included 3,893 participants who were followed up to 52 weeks. Shoulder-specific ET (Mean difference (MD) = -2.1; 95% confidence interval (CI) = -3.5 to -0.7) or in combination with electro-physical agents (MD = -2.5; 95% CI = -4.2 to -0.7), injections (MD = -2.4; 95% CI = -3.9 to—1.04) or manual therapy (MD = -2.3; 95% CI = -3.7 to -0.8) decreased pain compared to usual medical care. Trends with ROM and HRQL scores were seen; however, only Manual Therapy (MD = -12.7 and 95% CI = -24.4 to -1.0) achieved meaningfully important changes. Sensitivity analysis excluding studies with high risk of bias showed similar results, with exception of injections that did not reach significance (MD = -1.3; 95% CI = -4.3 to 1.7).

**Data Availability Statement:** All relevant data are within the paper and its Supporting Information files.

**Funding:** The author(s) received no specific funding for this work.

**Competing interests:** The authors have declared that no competing interests exist.

## Conclusion(s)

Shoulder-specific ET provided pain relief up to 52 weeks. Adjunct therapies to shoulder-specific ET added little value in reducing pain. The quality of evidence varied between moderate and very low.

## Introduction

Chronic shoulder pain is highly prevalent, with incidence rates ranging from 7.7 to 62 per 1000 persons per year and community prevalence ranging from 0.67 to 55.2% worldwide [1]. It significantly impacts patients' quality of life including health-related quality of life (HRQL) and health care utilization [2]. In Canada, for example, treating chronic shoulder pain due to rotator cuff tears has an estimated cost between Can\$43million and Can\$101 million annually [3]. Evidence-based guidelines on effective management strategies for chronic shoulder pain are unclear due to high heterogeneity in treatment approaches, patient populations and study methodologies used [4]. Current clinical recommendations suggest a trial of conservative management (i.e., physiotherapy, medications) followed by surgery when conservative management is ineffective for chronic shoulder pain [5]. Rehabilitation of chronic shoulder pain through exercise therapy (ET) appear to be effective in pain relief and function gains, leading to increased participation in daily activities and better HRQL [6].

Following usual medical care (information, recommendations, and medical or pharmaceutical therapy as needed), exercise therapy (ET) is frequently a treatment choice when managing shoulder pain, yet evidence on its efficacy to expedite recovery is inconsistent [4,7]. Moreover, the value of adjunct therapies such as manual therapy, electro-physical agents, medications, and injections with ET is currently inconsistent. Although ET for shoulder pain is supported by 10 systematic reviews [7–16], only two [7,12] had strong recommendations for the use of ET. While findings from seven systematic reviews support using a combination of manual therapy (MT) and ET for pain relief and functional improvement, others state inconclusive evidence to support a combination of MT and ET. Inconclusive findings are also reported with the use of corticosteroid injections [15,17]. Shoulder diagnosis, ET definitions, follow-up time are highly variable among these systematic reviews and limit comparison.

Current systematic reviews on the benefits of conservative management for shoulder pain are mostly based on either descriptive analysis or limited meta-analysis, due to data heterogeneity with variability seen with outcomes, timelines, treatment length, follow-ups, and case definitions. A persuasive concern with many of these reviews is that ET was evaluated as one general approach, although ET consists of several different approaches including shoulder-specific strengthening and ROM exercises with/without scapular exercises to non-specific shoulder exercises such as postural and functional exercises. While the effectiveness of different types of ET has been evaluated with small clinical samples and systematic reviews, the different types of ET has yet to have head-to-head comparisons. Within a clinical context, ET is not always used alone but rather with adjunct therapy. Using a network meta-analysis, this study combined both direct and indirect evidence across studies on the effectiveness of ET with/without adjunct therapies compared to usual medical care for adults with chronic shoulder pain.

## Material and methods

This network meta-analysis (NMA) is registered in the PROSPERO database (CRD 4201935093). Initially, the protocol published at PROSPERO stated that a meta-analysis was

planned; however, we amended the protocol to include a network meta-analysis instead to enable the use of both direct and indirect evidence. We also amended the protocol to add the following inclusion criteria: "At least 6 weeks follow-up" and "More than 3 months of symptoms (chronic)". Such criteria were important to better define the population being studied and the changes were made before the review started. Preferred Reporting Items for Systematic Reviews and Meta-Analyses (PRISMA) (S1 Appendix) extension statement for network meta-analyses was followed [18].

## Eligibility criteria

This NMA included randomized or quasi-randomized control trials comparing ET with or without adjunct therapies in adult participants (aged 18 years or older) with shoulder pain for at least 3 months. At least one of the comparative groups needed to have ET as an intervention and follow-up time needed to be at least 6 weeks to detect true effect of ET. We excluded participants with previous surgery to the affected shoulder, history of shoulder dislocation, inflammatory disease, adhesive capsulitis (Frozen shoulder), scapular dyskinesis, major shoulder joint trauma, infection, avascular necrosis or neuropathy, or concomitant neck pathology. Studies not in English language, including fewer than 20 participants in the cohort, or examining holistic treatments were also excluded.

## Information sources and search

A research health sciences librarian developed and conducted a systematic search of the following databases up to May/2022 and limited to English language: MEDLINE, Embase, CINAHL, Sportdiscus, CENTRAL, Conference Proceedings Citation Index- Science (CPCI-S), clinicaltrials.gov, and association websites (Canadian Academy of Sport and Exercise Medicine, Canadian Athletic Therapists Association, Canadian Physiotherapy Association, College of Family Physicians of Canada–Sport & Exercise Committee, Exercise is Medicine Canada, Ontario Athletic Therapist Association, Ontario Medical Association–Section on Sport & Exercise Medicine, Sport Physiotherapy Canada). Search strategy can be found in (S2 Appendix).

## Study selection

Two independent reviewers (AS, CL) used Covidence[TM] [19], for title, abstract, full text screening and data extraction. Disagreement of article inclusion was resolved through consensus between reviewers or through third party adjudication if the reviewers did not arrive at consensus. Study authors were contacted if further clarifications regarding study methods and/or results were needed.

## Data extraction

Two independent reviewers (AS, CL) extracted the following data from eligible studies: demographics (number participants, age, sex, year, country, and diagnosis), interventions (type, duration, retention, maximum follow-up time) and outcomes (Pain, ROM, HRQL). All outcomes were extracted for the following timelines: post-intervention (first follow-up once intervention was completed) and longest follow-up (last study follow-up). If outcome information was unclear in the manuscript, we contacted the authors for clarifications.

## Quality and publication bias assessment

Two independent reviewers (AS, CM) used the Revised Cochrane risk-of-bias tool for randomized trials (RoB2) to assess the quality of each study [20]. Overall bias scores used the following criteria: **low risk of bias** (all domains were low), **some concerns** (at least one domain had some concerns, but none had high risk of bias) and **high risk of bias** (at least one domain was high or had some concerns in multiple domains that decreased the confidence in the result) [20].

Risk Of Bias due to Missing Evidence in Network meta-analysis (ROB-MEN) [21] assessed risk of bias due to missing evidence (publication bias) for all included pairwise comparisons. This assessment considered: 1. the contribution of direct comparisons to the network meta-analysis estimates, 2. the potential presence of small-study effects, and 3. any bias from unobserved comparisons. The automatized tool then assigned a level of low risk, some concerns, or high risk of bias [21].

## Certainty of evidence

Reviewers used the Confidence in Network Meta-Analysis (CINeMA) [22,23] to assess the certainty of evidence for all outcomes considering the following domains: within-study bias, reporting/publication bias, indirectness, imprecision, heterogeneity, and incoherence. Final judgment summary across all domains were based on GRADE framework [22,23]. Reviewers took into consideration that domains may be interconnected and followed CINeMA guidelines for judgment to avoid downgrading the overall level of confidence more than once for related concerns. Indirectness and incoherence were considered correlated and heterogeneity, imprecision, within-study bias and reporting bias were considered correlated.

This NMA included the assumptions of consistency and transitivity. CINeMA assessed consistency though the design-by-treatment test and by separating indirect from direct evidence (SIDE test) using the R netmeta package$^{TM}$. For the transitivity assumption, CINeMA considers indirectness through the distribution of potential effect modifiers, and statistical incoherence through the SIDE test [22,23].

## Outcomes

Based on the literature, we anticipated ET with or without adjunct therapies to have an impact on shoulder pain, ROM, strength, and HRQL. Such domains are clinically important to both the patients and the clinicians to access effectiveness of therapies [24].

Because shoulder-specific pain was measured through several pain scales, all scores were transformed to a scale from 0 (no pain) to 10 (worst pain) for comparison. We considered a difference of 20% to be clinically important [25].

Shoulder abduction and external rotation are the most restricted ROM in patients with chronic shoulder pain and dysfunction. Even though a minimal clinically important difference has yet to be established for chronic shoulder pain population, based on the current literature a difference of 10 degrees was considered clinically important for this NMA [26].

Disease-specific HRQL measures such as Shoulder Pain and Disability Index (SPADI), The Disabilities of the Arm, Shoulder and Hand (DASH), and Quick-DASH were included. A 10 points difference was considered clinically important [27–30].

## Data synthesis and analysis

Data synthesis pooled data for the outcomes of interest in the pre-specified groups, including mean or mean differences, standard deviations (SD) and/or 95% confidence intervals (Cis),

follow up time, number of included participants per group, demographics (age, gender), and exercises program characteristics (total duration, post-intervention and retention). Groups were classified as the following (S3 Appendix).

1. **Rotator Cuff and Scapula Exercise (RC+SCAP):** Participants allocated to this group were treated with an exercise program that targeted both rotator cuff and scapular muscles.

2. **Rotator Cuff Exercise (RC):** Participants allocated to this group were treated with an exercise program that targeted mostly rotator cuff muscles without focusing on scapula muscles.

3. **Non-Specific RC Exercises:** Participants allocated to this group were treated with an exercise program that did not target specifically RC muscles.

4. **Electro-physical agent (EPA) + Exercise Therapy (ET):** Participants allocated to this group used electro-physical modalities in addition to their exercise program. Modalities included electrotherapy (i.e. TENS, ultrasound, laser, IFC, microwave diathermy, and/or radial extracorporeal shock-wave), thermotherapy (cold and/or heat), and dry needling.

5. **Manual Therapy (MT) + ET:** Participants allocated to this group used manual therapy in addition to their exercise program. Manual therapy techniques could include any of the following: soft tissue massage, joint mobilization (i.e. Glenohumeral, scapula, acromioclavicular, sternoclavicular, cervical and/ or thoracic), and/or manual compression of trigger points.

6. **Injections + ET:** Participants allocated to this group used injections (i.e. corticosteroid, prolotherapy, platelet-rich plasma) in addition to their exercise program.

   **Usual Care:** Participants allocated to this group saw their family physician who gave them information, recommendations, and medical or pharmaceutical therapy as needed. Patients followed a wait-and-see approach and re-consulted with their family physician if symptoms persisted for further evaluation. We also included participants that received no treatment during the study in this group.

## Statistical analysis

Data analysis combined direct and indirect comparisons in a Frequentist hierarchical model. Data was combined using a random-effects model and included information from all studies. Relative effects (Mean differences) and a common heterogeneity parameter ($\tau^2$) using R Net-Meta package™ were estimated using CINeMA™ for all outcomes. Assessment of the agreement of the various sources of evidence was calculated using the design-by-treatment test and by separating indirect from direct evidence (SIDE test) using the R netmeta package™ in CINeMA™ [22,23].

CINeMA used the flow decomposition method [22,23] to calculate the contribution matrix. Contribution matrix included the percentage contribution of information from each study and each direct comparison to the estimation of each relative effect. Contribution matrix was used in the evaluation of contribution of within-study bias and indirectness to the confidence in the results.

NMA plots visually showed direct comparisons through edges. Nodes size represented the number of participants assigned to each intervention and node color represented ROB as described above. We imputed baseline standard deviation (SD) values when presented with mean differences from the baseline, but without a correspondent SD. Publication Bias used ROBMEN™ [21] to assess the risk of bias due to missing evidence for all possible pairwise

comparisons in the network. Sensitivity analyses (excluding studies at overall high risk of bias) controlled for residual bias. The strength of evidence was measured by a synthesis of each outcome using the framework described by Salanti and colleagues [31] and implemented using the CINeMA[TM] [22,23] which allowed confidence in the results to be graded as high, moderate, low, and very low.

## Results

Literature search identified 16,641 citations, of which 5,678 duplicates and 10,052 were excluded. Of the 911 full-text studies reviewed, 54 studies were included [32–86] in the Network Meta-analysis (Fig 1). 4 articles [60,61,73,74] were from 2 studies and, in the analysis, we accounted as 2 studies instead of 4. 22 (43.31%) studies received research grant funds, 3 (5.77%) received industry funds, 5 (9.62%) stated no funds and 22 (42.31%) had no information regarding funds. The majority of included studies were from Europe (21.40.38%) and the remaining from Asia, (18, 34.6%), South America (5 (9.62%), North America (4,7.69%), and Australia (4 (7.69%). 22 out 52 studies had published protocols available [32,33,36,37,40,43,44,47,53,58,61,65,68,69,71,72,74–76,78,80,82].

### Characteristics of the included studies

Of the 3,893 participants, the mean age was 51.26 years (SD: 7.55) with slightly over half being female (2,053 (52.7%)). The primary diagnosis was rotator cuff related shoulder pain (79%) with the remaining diagnosed as unspecified shoulder pain (21%). 6 interventions were compared to usual care. The mean intervention duration was 7.09 weeks (SD = 3.67).

### Quality and publication bias assessment

Overall risk of bias assessment (S4 Appendix) for **pain** found 19 studies at high risk [34,38,39,41,43,45–48,57–59,64,66,70,78,79,81,84], 21 with some concerns

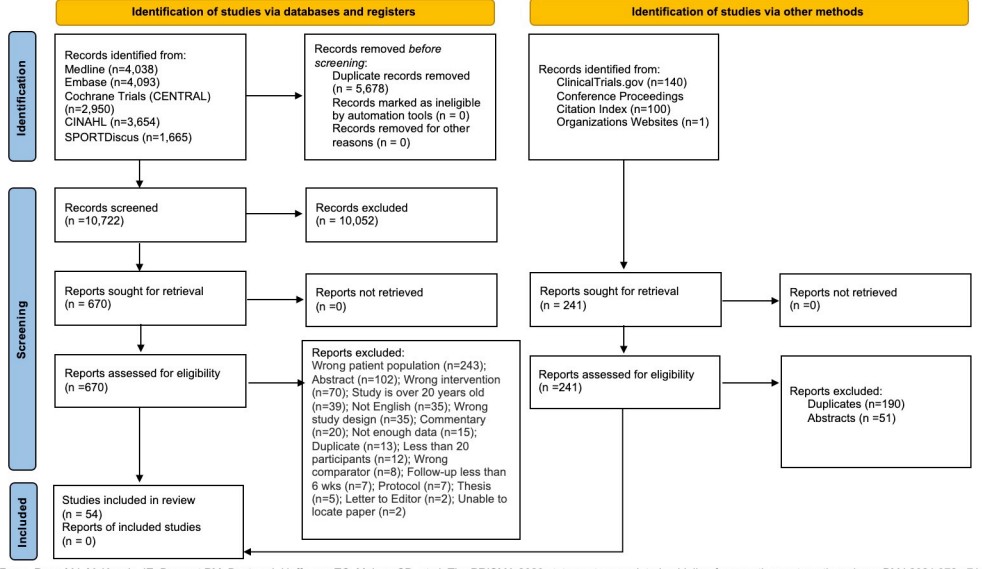

*From:* Page MJ, McKenzie JE, Bossuyt PM, Boutron I, Hoffmann TC, Mulrow CD, et al. The PRISMA 2020 statement: an updated guideline for reporting systematic reviews. BMJ 2021;372:n71. doi: 10.1136/bmj.n71. For more information, visit: http://www.prisma-statement.org/

**Fig 1. PRISMA 2020 flow diagram for new systematic reviews which included searches of databases, registers and other sources.**

[33,35,37,40,42,44,51,52,54,56,60–62,65,68,77,80,82,85] and 4 at low risk [53,71,73,74]. For **ABD ROM,** 9 studies had high risk [34,41,57,59,62,70,75,81,84] and 6 had some concerns [35,40,49,56,77,85]. **ER ROM** had 12 studies with high risk [34,38,39,41,48,57,59,62,70,75,81,84] and 5 with some concerns [35,40,56,63,77]. **SPADI** consisted of 8 studies at high risk [32,38,43,59,75,76,83,84], 13 with some concerns [35,42,51,52,60,61,65,67,72,80,82,85,86] and 3 with low risk of bias [69,73,74]. Finally, DASH had 8 studies at high risk [32,34,47,50,66,70,79,81], 7 with some concerns [33,36,37,44,51,52,68,80] and 2 at low risk [53,71].

ROB-MEN risk of bias due to missing evidence showed some concerns for EPA+ET, RC +SCAP and Non-specific RC exercises compared to usual care for pain. For ROM, some concerns were seen for RC and EPA+ET. Finally, SPADI had some concerns with RC+SCAP. (S5 Appendix)

## Outcomes

**Pain.** All ET approaches showed large significant pain relief when compared to usual medical care post-intervention: **EPA+ET** (MD = -2.5; 95% CI = -4.2 to -0.7), **Injections+ET** (MD = -2.4; 95% CI = -3.9 to -1.04), **MT+ET** (MD = -2.3; 95% CI = -3.7 to -0.8), and **RC +SCAP** (MD = -2.1; 95% confidence interval (CI) = -3.5 to -0.7) (Table 1). When studies with high RoB were removed, the sensitivity analysis (S6 Appendix), however, showed that injections lost both statistical and clinical significance (MD = -1.28; 95% CI = -4.28 to 1.73). SIDE test showed no major concerns with inconsistency (P>0.05; S6 Appendix).

Up to 52 weeks post-intervention (longest follow-up), pain relief was retained for **EPA+ET** (MD = -2.6 and 95% CI = -4.0 to -1.2), **Injections+ET** (MD = -2.9 and 95% CI = -4.6 to -1.2), **MT+ET** (MD = -2.3 and 95% CI = -3.6 to -0.9), **RC** (MD = -1.7 and 95% CI = -3.3 to -0.1) and **RC+SCAP** (MD = -2.1 and 95% CI = -3.5 to -0.8). However, once again adding injections to ET did not show significant or clinically important pain relief when excluding high RoB studies. Confidence in the results varied between moderate to very low (Table 2). SIDE test showed no major concerns with inconsistency (P>0.05; S6 Appendix).

## Shoulder ROM and HRQL

ROM (ER; ABD) included 917 and 894 participants post-intervention. The average ER for the shoulder was 67.38 degrees (Min 36.5 to Max 95) [34,35,38–41,46,56,57,59,62,70,75,77,81,84] and for abduction was 135.8 degrees (Min 9.33 to Max 179.5). [34,35,40,41,49,57,59,62,70,75,77,81,84,85] Shoulder-specific HRQL (SPADI, DASH) included 2,375 and 1,154 participants respectively post-intervention. SPADI [32,35,38,42,43,49–52,59–61,65,67,69,72–74,76,80,82–86] average was 30.22 points (Min 10.1 to Max 61.4) and for DASH [32,34,36,37,44,47,50–53,66,68,70,71,77,80,81] was 26.46 points (Min 9.3 to Max 51.35). There was a trend in improving ROM (ER; ABD) and HRQL (SPADI, DASH) when

**Table 1. Mean differences (95%CI) for pain relief post-intervention.** Statistically significant differences are in red.

| EPA | -0.042 (-1.275, 1.191) | -0.200 (-1.186, 0.785) | -1.174 (-2.810, 0.463) | -0.955 (-2.189, 0.278) | -0.363 (-1.280, 0.554) | **-2.459 (-3.875, -1.044)** |
|---|---|---|---|---|---|---|
| 0.042 (-1.191, 1.275) | Injections | -0.159 (-1.558, 1.241) | -1.132 (-3.010, 0.747) | -0.914 (-2.446, 0.620) | -0.321 (-1.606, 0.964) | **-2.418 (-4.152, -0.684)** |
| 0.200 (-0.785, 1.186) | 0.159 (-1.241, 1.558) | Manual Therapy | -0.973 (-2.675, 0.729) | -0.755 (-2.183, 0.673) | -0.162 (-1.222, 0.897) | **-2.259 (-3.674, -0.844)** |
| 1.174 (-0.463, 2.810) | 1.132 (-0.747, 3.010) | 0.973 (-0.729, 2.675) | Non-specific RC exercises | 0.218 (-1.607, 2.043) | 0.811 (-0.606, 2.227) | -1.286 (-3.050, 0.478) |
| 0.955 (-0.278, 2.189) | 0.914 (-0.620, 2.446) | 0.755 (-0.673, 2.183) | -0.218 (-2.043, 1.607) | RC | 0.592 (-0.634, 1.819) | -1.504 (-3.123, 0.114) |
| 0.363 (-0.554, 1.280) | 0.321 (-0.964, 1.606) | 0.162 (-0.897, 1.222) | -0.811 (-2.227, 0.606) | -0.592 (-1.819, 0.634) | RC+SCAP | **-2.097 (-3.491, -0.702)** |
| 2.459 (1.044, 3.875) | 2.418 (0.684, 4.152) | 2.259 (0.844, 3.674) | 1.286 (-0.478, 3.050) | 1.504 (-0.114, 3.123) | 2.097 (0.702, 3.491) | Usual Medical Care |

**Table 2. Pain Confidence in the results.**

| Compari-son | Number of studi-es | Within-study bias | Report-ing bias | Indirectness | Impreci-sion | Heteroge-neity | Incohe-rence | Confi-dence rating | Reason(s) for downgrading |
|---|---|---|---|---|---|---|---|---|---|
| | | | | **MIXED EVIDENCE** | | | | | |
| EPA+ET: Injections+ET | 3 | Major concerns | Low risk | Some concerns | No concerns | Major concerns | No concerns | Very low | 2 levels for major concerns with heterogeneity and within-study bias. 1 level for some concerns with indirectness. |
| EPA+ET:MT | 6 | Some concerns | Low risk | Some concerns | No concerns | Major concerns | No concerns | Very low | 2 levels for major concerns with heterogeneity and some concerns within-study bias. 1 level for some concerns with indirectness. |
| EPA+ET:RC | 3 | Some concerns | Low risk | Some concerns | Some concerns | Some concerns | No concerns | Low | 1 level for some concerns with heterogeneity, imprecision and within-study bias. 1 level for some concerns with indirectness. |
| EPA+ET:RC +SCAP | 6 | Some concerns | Low risk | Some concerns | No concerns | Major concerns | No concerns | Very low | 2 levels for major concerns with heterogeneity and some concerns within-study bias. 1 level for some concerns with indirectness. |
| EPA+ET:Usual Medical Care | 1 | Some concerns | Some concerns | Some concerns | No concerns | Some concerns | No concerns | Low | 1 level for some concerns with heterogeneity, reporting bias and within-study bias. 1 level for some concerns with indirectness. |
| Injections+ET: MT | 1 | Major concerns | Low risk | Some concerns | No concerns | Major concerns | No concerns | Very low | 2 levels for major concerns with heterogeneity and within-study bias. 1 level for some concerns with indirectness. |
| Injections+ET:RC | 1 | Major concerns | Some concerns | Some concerns | Some concerns | Some concerns | No concerns | Very low | 2 levels for major concerns with within-study bias and some concerns with heterogeneity and reporting bias. 1 level for some concerns with indirectness. |
| Injections+ET: RC+SCAP | 2 | Major concerns | Low risk | Some concerns | No concerns | Major concerns | No concerns | Very low | 2 levels for major concerns with heterogeneity and within-study bias. 1 level for some concerns with indirectness. |
| MT: RC+SCAP | 4 | Some concerns | Low risk | Some concerns | No concerns | Major concerns | No concerns | Very low | 2 levels for major concerns with heterogeneity and some concerns within-study bias. 1 level for some concerns with indirectness. |
| MT:Usual Medical Care | 2 | Some concerns | Low risk | Some concerns | No concerns | Some concerns | No concerns | Low | 1 level for some concerns with heterogeneity and within-study bias. 1 level for some concerns with indirectness. |
| Non-specific RC exercises: RC+SCAP | 4 | Some concerns | Low risk | No concerns | Some concerns | Some concerns | No concerns | Moderate | 1 level for some concerns with heterogeneity, imprecision and within-study bias. |
| Non-specific RC exercises:Usual Medical Care | 1 | Major concerns | Some concerns | No concerns | Some concerns | Some concerns | No concerns | Low | 2 levels for major concerns with within-study bias and some concerns with heterogeneity, imprecision and reporting bias. |

*(Continued)*

**Table 2.** (Continued)

| Compari-son | Number of studi-es | Within-study bias | Report-ing bias | Indirectness | Impreci-sion | Heteroge-neity | Incohe-rence | Confi-dence rating | Reason(s) for downgrading |
|---|---|---|---|---|---|---|---|---|---|
| RC:RC+SCAP | 3 | Major concerns | Low risk | Some concerns | No concerns | Major concerns | No concerns | Very low | 2 levels for major concerns with within-study bias and heterogeneity. 1 level for some concerns with indirectness. |
| RC:Usual Medical Care | 1 | Some concerns | Low risk | No concerns | Some concerns | No concerns | No concerns | Moderate | 1 level some concerns with imprecision and within-study bias. |
| RC+SCAP:Usual Medical Care | 1 | Some concerns | Some concerns | No concerns | No concerns | Some concerns | Some concerns | Moderate | 1 level some concerns with imprecision, reporting bias, heterogeneity, incoherence and within-study bias. |
| INDIRECT EVIDENCE | | | | | | | | | |
| EPA+ET:Non-specific RC exercises | 0 | Some concerns | Low risk | No concerns | Some concerns | Some concerns | No concerns | Moderate | 1 level for some concerns with imprecision, heterogeneity, and within-study bias. |
| Injections+ET: Non-specific RC exercises | 0 | Major concerns | Low risk | No concerns | Some concerns | Some concerns | No concerns | Low | 2 levels for major concerns with within-study bias and some concerns with imprecision and heterogeneity. |
| Injections+ET: Usual Medical Care | 0 | Major concerns | Low risk | Some concerns | No concerns | Some concerns | No concerns | Very low | 2 levels for major concerns with within-study bias and some concerns with heterogeneity. 1 level for some concerns with indirectness. |
| MT:Non-specific RC exercises | 0 | Some concerns | Low risk | No concerns | Some concerns | Some concerns | No concerns | Moderate | 1 level for some concerns with imprecision, heterogeneity, and within-study bias. |
| MT:RC | 0 | Some concerns | Low risk | Some concerns | Some concerns | Some concerns | No concerns | Low | 1 level for some concerns with heterogeneity, imprecision and within-study bias. 1 level for some concerns with indirectness. |
| Non-specific RC exercises:RC | 0 | Some concerns | Low risk | No concerns | Some concerns | Some concerns | No concerns | Moderate | 1 level for some concerns with imprecision, heterogeneity, and within-study bias. |

compared to usual medical care; however, none achieved statistical and clinically important significant improvements (Tables 3–6). When high RoB studies were excluded, the improving trend was not seen with Injections+ET (SPADI; DASH) and non-specific RC exercises

**Table 3. Mean differences for the post-intervention ROM_ER.**

| | | | | | |
|---|---|---|---|---|---|
| **EPA** | 2.205 (-7.421, 11.830) | -0.441 (-10.981, 10.099) | 1.798 (-13.672, 17.268) | 1.678 (-6.449, 9.805) | 13.998 (-13.972, 41.968) |
| -2.205 (-11.830, 7.421) | **Injections** | -2.646 (-16.221, 10.930) | -0.406 (-17.760, 16.948) | -0.526 (-10.658, 9.605) | 11.793 (-17.261, 40.848) |
| 0.441 (-10.099, 10.981) | 2.646 (-10.930, 16.221) | **Manual Therapy** | 2.239 (-15.795, 20.273) | 2.119 (-9.340, 13.579) | 14.439 (-15.026, 43.905) |
| -1.798 (-17.268, 13.672) | 0.406 (-16.948, 17.760) | -2.239 (-20.273, 15.795) | **RC** | -0.120 (-15.376, 15.136) | 12.200 (-11.102, 35.502) |
| -1.678 (-9.805, 6.449) | 0.526 (-9.605, 10.658) | -2.119 (-13.579, 9.340) | 0.120 (-15.136, 15.376) | **RC+SCAP** | 12.320 (-15.532, 40.172) |
| -13.998 (-41.968, 13.972) | -11.793 (-40.848, 17.261) | -14.439 (-43.905, 15.026) | -12.200 (-35.502, 11.102) | -12.320 (-40.172, 15.532) | **Usual Medical Care** |

**Table 4. Mean differences for the post-intervention ROM_ABD.**

| EPA | 17.920 (-5.373, 41.213) | -0.272 (-26.125, 25.581) | 6.153 (-21.782, 34.089) | 3.989 (-14.827, 22.805) | 15.853 (-35.645, 67.352) |
|---|---|---|---|---|---|
| -17.920 (-41.213, 5.373) | Injections | -18.192 (-49.442, 13.058) | -11.767 (-45.119, 21.585) | -13.932 (-36.812, 8.949) | -2.067 (-56.694, 52.560) |
| 0.272 (-25.581, 26.125) | 18.192 (-13.058, 49.442) | Manual Therapy | 6.425 (-22.798, 35.649) | 4.261 (-19.772, 28.293) | 16.125 (-36.083, 68.334) |
| -6.153 (-34.089, 21.782) | 11.767 (-21.585, 45.119) | -6.425 (-35.649, 22.798) | RC | -2.165 (-29.271, 24.941) | 9.700 (-33.564, 52.964) |
| -3.989 (-22.805, 14.827) | 13.932 (-8.949, 36.812) | -4.261 (-28.293, 19.772) | 2.165 (-24.941, 29.271) | RC+SCAP | 11.865 (-39.189, 62.919) |
| -15.853 (-67.352, 35.645) | 2.067 (-52.560, 56.694) | -16.125 (-68.334, 36.083) | -9.700 (-52.964, 33.564) | -11.865 (-62.919, 39.189) | Usual Medical Care |

**Table 5. Mean differences for the post-intervention SPADI.**

| EPA | -0.205 (-15.169, 14.759) | -5.132 (-15.930, 5.666) | 7.539 (-13.711, 28.790) | -6.452 (-22.131, 9.227) | -4.491 (-14.528, 5.545) | -14.390 (-31.776, 2.995) |
|---|---|---|---|---|---|---|
| 0.205 (-14.759, 15.169) | Injections | -4.927 (-19.093, 9.239) | 7.744 (-17.382, 32.871) | -6.247 (-25.779, 13.285) | -4.286 (-19.228, 10.655) | -14.185 (-34.711, 6.341) |
| 5.132 (-5.666, 15.930) | 4.927 (-9.239, 19.093) | Manual Therapy | 12.671 (-10.192, 35.534) | -1.320 (-16.804, 14.164) | 0.640 (-9.987, 11.268) | -9.258 (-25.721, 7.204) |
| -7.539 (-28.790, 13.711) | -7.744 (-32.871, 17.382) | -12.671 (-35.534, 10.192) | Non-specific RC exercises | -13.992 (-39.447, 11.464) | -12.031 (-33.584, 9.523) | -21.930 (-48.328, 4.468) |
| 6.452 (-9.227, 22.131) | 6.247 (-13.285, 25.779) | 1.320 (-14.164, 16.804) | 13.992 (-11.464, 39.447) | RC | 1.961 (-13.307, 17.229) | -7.938 (-23.403, 7.527) |
| 4.491 (-5.545, 14.528) | 4.286 (-10.655, 19.228) | -0.640 (-11.268, 9.987) | 12.031 (-9.523, 33.584) | -1.961 (-17.229, 13.307) | RC+SCAP | -9.899 (-26.396, 6.598) |
| 14.390 (-2.995, 31.776) | 14.185 (-6.341, 34.711) | 9.258 (-7.204, 25.721) | 21.930 (-4.468, 48.328) | 7.938 (-7.527, 23.403) | 9.899 (-6.598, 26.396) | Usual Medical Care |

(DASH) (S6 Appendix). Post-intervention trends were retained up to 52 weeks with exception of MT+ET that showed a significant and clinically important improvement in DASH scores (MD = -12.7 and 95% CI = -24.4 to -1.02); however, such improvement disappeared when excluding high RoB studies (MD = -7.7 and 95% CI = -21.1 to 5.7). Confidence in the results varied between moderate to very low (S7 Appendix). SIDE test showed *some* to *major* concerns with inconsistency for DASH retention, mainly in the indirect comparisons (P = 0.047); S6 Appendix). For all other outcomes, SIDE test showed no *major* concerns with inconsistency (P>0.05; S6 Appendix).

**Table 6. Mean differences for the post-intervention DASH.**

| EPA | -7.281 (-17.034, 2.472) | -1.028 (-10.480, 8.423) | -10.511 (-24.302, 3.280) | -8.358 (-20.422, 3.705) | -2.159 (-12.180, 7.861) | -13.057 (-27.963, 1.850) |
|---|---|---|---|---|---|---|
| 7.281 (-2.472, 17.034) | Injections | 6.253 (-2.234, 14.739) | -3.230 (-16.598, 10.138) | -1.078 (-12.894, 10.738) | 5.122 (-4.308, 14.551) | -5.776 (-20.264, 8.712) |
| 1.028 (-8.423, 10.480) | -6.253 (-14.739, 2.234) | Manual Therapy | -9.482 (-21.951, 2.986) | -7.330 (-19.702, 5.041) | -1.131 (-9.235, 6.974) | -12.028 (-25.276, 1.220) |
| 10.511 (-3.280, 24.302) | 3.230 (-10.138, 16.598) | 9.482 (-2.986, 21.951) | Non-specific RC exercises | 2.152 (-14.191, 18.495) | 8.352 (-1.124, 17.827) | -2.546 (-20.230, 15.139) |
| 8.358 (-3.705, 20.422) | 1.078 (-10.738, 12.894) | 7.330 (-5.041, 19.702) | -2.152 (-18.495, 14.191) | RC | 6.199 (-7.116, 19.515) | -4.698 (-18.619, 9.222) |
| 2.159 (-7.861, 12.180) | -5.122 (-14.551, 4.308) | 1.131 (-6.974, 9.235) | -8.352 (-17.827, 1.124) | -6.199 (-19.515, 7.116) | RC+SCAP | -10.898 (-25.829, 4.034) |
| 13.057 (-1.850, 27.963) | 5.776 (-8.712, 20.264) | 12.028 (-1.220, 25.276) | 2.546 (-15.139, 20.230) | 4.698 (-9.222, 18.619) | 10.898 (-4.034, 25.829) | Usual Medical Care |

## Discussion

Findings from this NMA were that shoulder-specific strengthening along with scapular exercises and ROM exercises are more effective in providing pain relief for chronic shoulder pain than usual medical care. Pain relief can last up to 52 weeks following an average of 7.09 weeks ET program. Evidence shows that targeting specifically shoulder muscles improves shoulder biomechanics, leading to better movement patterns that decreases shoulder impingement and allows shoulder healing [87]. A recent RCT [88] showed that a 12-weeks supervised rehabilitation program using shoulder-specific exercises with the addition of scapular retraction exercises was effective in decreasing patients' pain and improving HRQL. However, another RCT [89] stated that adding 12-week ET (shoulder-specific or functional exercises) to formal shoulder pain education did not result in further benefits to the patients. Dube's (2023) [89] study had a well-defined education group including information on shoulder (anatomy and function), pain mechanism, pain management and activity modification. Moreover, participants watched educational videos on shoulder pain/function, chronic pain, stress, and the importance of healthy habits (sleep, eating and physical exercise). Usual care in this NMA may or may not have included an education component as part of their intervention and the content of education intervention varied among studies. Furthermore, it is important to take into account that education interventions are highly correlated with patients' levels of education and their ability to understand and implement the recommendations [90,91]. Exercise recommendations were also part of the education component in Dube (2023) [89] study and may also have contributed to their findings.

Usual medical care frequently relies on the use of pharmaceutical management including NSAIDS and corticosteroid injections to reduce pain by decreasing the inflammatory process commonly seen in patients with chronic shoulder pain; however, the evidence is of low quality [15]. Even though a systematic review showed that both NSAIDS (SMD of −0.29, 95% CI −0.53 to −0.05) and corticosteroid injections (SMD −0.65, 95% CI −1.04 to −0.26) were more effective than no treatment, included studies were of low quality and it remained unclear how pharmaceutical management compared to ET [15]. This NMA adds value to the current literature since it shows that shoulder-specific strengthening and ROM exercises including scapular exercises provides long-lasting pain relief for chronic shoulder pain compared to usual medical care. It also included studies that had at least 6 weeks follow-up, an important factor to detect true effect of ET. Finally, this NMA was not restricted to a specific shoulder diagnosis, rather it engloped the most common diagnosis of shoulder pain under the umbrella of rotator cuff related shoulder pain as well as unspecified shoulder pain that better reflects the population seen under current primary care.

Shoulder pain is the primary reason people seek medical treatment, since pain impacts patients both physically and emotionally [24]. Adding adjunct therapies to ET added little value when compared to shoulder-specific ET in reducing pain. We found that the addition of injections to ET lost both statistical and clinical importance compared to usual care which typically included medication. A systematic review [16] however showed that injections (SMD −0.65, 95% CI −1.04 to −0.26) were more effective than no treatment. Injections may be effective in reducing pain by decreasing the inflammatory process commonly seen in patients with chronic shoulder pain; however, the evidence is of low quality [16].

### Strengths and limitations

This NMA has several strengths. Inclusion of RCTs studies ensured conclusions were based on best available evidence. Exclusion of studies with less than 6 weeks follow-ups enabled reliable assessment of the effect of ET with or without the addition of adjunct therapies. To the best of

our knowledge, this is the first NMA that classified ET interventions taking into consideration targeted muscles (RC muscles, RC and scapula muscles or non-specific shoulder muscles) as well as did not focus the interventions to a specific diagnosis. The large sample size (3,893) increased the power of the results. We also considered the effects of the interventions immediately post intervention and at the longest follow-up, enabling conclusions regarding intervention effect and retention.

This NMA is limited by the quality of included studies, since most studies were considered as moderate to high risk of bias. Definition of ET depended on exercises strategies; however, authors are physical therapists with specialty training in shoulder treatments increasing the reliability of definitions [4]. Adjunct therapies were considered in combination with ET and not as stand-alone interventions, limiting the conclusions regarding their effectiveness on their own. Usual care group included variable approaches, including advice, wait-and-see and potential use of pain medications; however, in current practice this is a very common pattern [92,93]. The diagnosis criteria were variable among studies, but we used rotator cuff related shoulder pain or undefined shoulder pain terms to address this concern. We were unable to compile strength data, an important outcome to reflect ET effectiveness, due to inconsistency in measurement methods. The classification of groups in this NMA limits the ability to effectively assess targeted treatment effects of individual interventions that may have different mechanisms and effects if considered separately. In some cases, the small number of studies prevented the analysis of specific interventions such as injections and dry needling. The authors considered the purpose of each intervention to group interventions with similar approaches so there was less variability within groups.

## Implications for clinical practice

Shoulder pain has deleterious impact on functional activities, overall health status and is associated with increased health care utilization and associated costs [3]. Health care providers need to take into consideration not only the best treatments available to treat shoulder pain but also the costs associated with each treatment. In this NMA, ET targeting shoulder muscles decreased meaningfully shoulder pain and the addition of adjunct therapies had questionable value. On the other hand, the effect of ET and adjunct therapies on shoulder ROM and HRQL did not show significant differences. Since pain is the major reason patients seek treatment [93], we advocate that ET be considered as the first line of treatment when dealing with chronic shoulder pain.

## Implications for future research

Pulling data for this NMA highlighted important barriers that need to be addressed in future trials. First, most included studies lacked published protocols, limiting the ability to judge their findings and increasing the risk of bias of included trials. Secondly, replicability and quality of studies requires detailed information on study methodology [94]. Most studies included in this NMA included general descriptions of interventions limiting the ability to properly combine information into groups and to replicate interventions in real-life clinical settings. Thirdly, the length of interventions varied between 2 to 16 weeks, yet ET requires at least 12-weeks to decrease pain and increase function [94]. Finally, even though strength is an important outcome when assessing the effectiveness of ET [95,96], it has been poorly reported and not feasible to synthesize the results in this NMA. Future studies need to address these barriers to increase confidence in the results and facilitate the implementation of effective interventions in clinical settings. These findings need be interpreted with caution, given the quality of evidence.

## Conclusion

Compared to usual care, shoulder-specific ET including scapular exercises are more effective in decreasing pain and maintaining pain relief. Adding adjunct therapies to ET resulted in little pain relief when compared to shoulder-specific ET and usual care. Although augmenting ET with MT had clinically important effects on health status, such effects were not seen when low quality studies were removed. Future studies need to consider important barriers such as having published protocols, including detailed information on study methodology and considering intervention lengths and responsive outcomes.

## Supporting information

**S1 Appendix. This is the PRISMA 2020 checklist.**
(DOCX)

**S2 Appendix. This is the search strattegy.**
(DOCX)

**S3 Appendix. This is the description of interventions by study.**
(DOCX)

**S4 Appendix. This is the risk of bias legend and table.**
(DOCX)

**S5 Appendix. This is the publication bias tables by outcomes.**
(DOCX)

**S6 Appendix. This is the post-intervention sensitivity analysis tables.**
(DOCX)

**S7 Appendix. This is the confidence in results tables tables.**
(DOCX)

## Acknowledgments

We thank Liza Bialy, MSc and Alberta Strategy for Patient-Oriented Research (SPOR) SUPPORT Unit Knowledge Translation Platform at University of Alberta for support with methodological design and data collection. We also thank Ben Vandermeer, MSc for support with data analysis.

## Author Contributions

**Conceptualization:** Anelise Silveira, Lauren Beaupre, Judy Chepeha, Allyson Jones.

**Data curation:** Anelise Silveira, Camila Lima.

**Formal analysis:** Anelise Silveira, Camila Lima.

**Methodology:** Anelise Silveira, Lauren Beaupre, Judy Chepeha, Allyson Jones.

**Project administration:** Anelise Silveira.

**Supervision:** Lauren Beaupre, Allyson Jones.

**Writing – original draft:** Anelise Silveira.

**Writing – review & editing:** Anelise Silveira, Camila Lima, Lauren Beaupre, Judy Chepeha, Allyson Jones.

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
