## [Decision Letter · Decision Letter 0]

17 Aug 2023

PONE-D-23-11628Shoulder specific exercise therapy is effective in reducing chronic shoulder pain: A network meta-analysisPLOS ONE

Dear Dr. Silveira,

Thank you for submitting your manuscript to PLOS ONE. After careful consideration, we feel that it has merit but does not fully meet PLOS ONE’s publication criteria as it currently stands. Therefore, we invite you to submit a revised version of the manuscript that addresses the points raised during the review process.

We look forward to receiving your revised manuscript.

Kind regards,

Juan Ignacio Guerrero-Henriquez, MSc

Academic Editor

PLOS ONE

Journal Requirements:

**Additional Editor Comments:**

Dear Anelise Silveira, I hope this email finds you well.

I am writing to you today as the editor of PlosOne to inform you that your manuscript, "Shoulder specific exercise therapy is effective in reducing chronic shoulder pain: A network meta-analysis", has been reviewed by three independent experts. The reviewers have raised some concerns that need to be addressed before the manuscript can be considered for publication.

I understand that revisions can be time-consuming, but I believe that your manuscript has the potential to be a valuable contribution to the field.

Thank you for your understanding and cooperation.

Sincerely,

Juan Guerrero Henríquez, MSc

Reviewers' comments:

Reviewer's Responses to Questions

**Comments to the Author**

1. Is the manuscript technically sound, and do the data support the conclusions?

Reviewer #1: Yes

Reviewer #2: Yes

Reviewer #3: Partly

2. Has the statistical analysis been performed appropriately and rigorously? 

Reviewer #1: No

Reviewer #2: Yes

Reviewer #3: Yes

3. Have the authors made all data underlying the findings in their manuscript fully available?

Reviewer #1: Yes

Reviewer #2: Yes

Reviewer #3: Yes

4. Is the manuscript presented in an intelligible fashion and written in standard English?

Reviewer #1: Yes

Reviewer #2: Yes

Reviewer #3: Yes

5. Review Comments to the Author

Reviewer #1: This paper is a well-constructed investigation into the evidence for shoulder-specific exercise therapy in the management of chronic shoulder pain, utilizing a network meta-analysis (NMA). The research goals are explicitly stated, and the research question is of clinical significance. Overall, the manuscript is clearly written, and the data presented in tables and graphs are highly informative.

1.On page 6, line 94, there seems to be a limitation regarding the eligibility criteria. The exclusion of participants with scapular dyskinesis may not fully represent the population of patients with chronic shoulder pain, many of whom present with this condition, especially those prescribed scapular exercises. Could you please clarify this point?

2.On page 8, line 137, it's important to note that NMA includes the assumption of consistency/transitivity. This aspect should be explicitly outlined. It would be beneficial if the authors could provide additional information on the NMA approach via CINeMA and the R Net-Meta package.

3.On page 9, line 165, the description of group classification is overly lengthy. Please consider revising for conciseness and clarity.

4.On page 10, lines 174 and 182, the classification of the EPA+ET group (electro-physical agents and exercise therapy) may not be entirely appropriate. Modalities such as electrotherapy, thermotherapy, and dry needling may have varying treatment effects on shoulder pain. Additionally, the combination of data for the injection+ET group may be questionable, given that steroids, prolotherapy, and PRP each have different mechanisms and effects. Would it be possible for the authors to perform a subgroup analysis or additional analysis to address these concerns?

5.Lastly, could you please provide the inconsistency results of the NMA?

Reviewer #2: Thank you for allowing me to review this manuscript which reviews the effectiveness of exercise-based therapeutic approaches as well as other commonly used interventions for chronic shoulder pain. The manuscript is well-written and provides quality information relevant to clinical decision-making for managing this prevalent musculoskeletal disorder.

Reviewer #3: Consider describing the objective of the research clearly, because there are two different objectives throughout the publication, this can be reviewed in line 5 and line 74.

Consider mentioning the definition of: "usual medical care" in the introduction and not only in the discussion on line 307.

Regarding the conclusion, consider answering the objective of the research by modifying the approach of the objectives mentioned above.

6. PLOS authors have the option to publish the peer review history of their article (what does this mean?). If published, this will include your full peer review and any attached files.

Reviewer #1: No

Reviewer #2: No

Reviewer #3: **Yes: **Diego Guerra-Rodriguez

---

## [Author Response · Author response to Decision Letter 0]

8 Sep 2023

September 8, 2023

Juan Ignacio Guerrero-Henriquez, MSc

Academic Editor

PLOS ONE

Dear Juan Ignacio Guerrero-Henriquez, Diego Guerra-Rodriguez and PLOS ONE reviewers,

I hope this letter finds you well. I would like to thank you all for your invaluable feedback and constructive comments on our manuscript titled " Shoulder specific exercise therapy is effective in reducing chronic shoulder pain: A network meta-analysis," which was submitted to PLOS ONE for review. Your thoughtful review has been instrumental in improving the quality and rigor of our research.

I am pleased to inform you that we have carefully considered each of your comments and have made the necessary revisions to address them. Below, I have summarized your major points of concern and provided our responses and actions taken:

Reviewer #1:

1. “On page 6, line 94, there seems to be a limitation regarding the eligibility criteria. The exclusion of participants with scapular dyskinesis may not fully represent the population of patients with chronic shoulder pain, many of whom present with this condition, especially those prescribed scapular exercises. Could you please clarify this point?” – Even though scapular dyskinesis may be related to chronic shoulder pain, scapular dyskinesis is also quite common in the asymptomatic population and many symptomatic patients do not necessarily present this condition. Therefore, we excluded scapular dyskinesis, since it could be a potential confounder to our findings. (1, 2)

2. “On page 8, line 137, it's important to note that NMA includes the assumption of consistency/transitivity. This aspect should be explicitly outlined. It would be beneficial if the authors could provide additional information on the NMA approach via CINeMA and the R Net-Meta package.” – Information added to lines 149-153.

3. “On page 9, line 165, the description of group classification is overly lengthy. Please consider revising for conciseness and clarity.” – Please see revised group classification (lines 177 and 179).

4. “On page 10, lines 174 and 182, the classification of the EPA+ET group (electro-physical agents and exercise therapy) may not be entirely appropriate. Modalities such as electrotherapy, thermotherapy, and dry needling may have varying treatment effects on shoulder pain. Additionally, the combination of data for the injection+ET group may be questionable, given that steroids, prolotherapy, and PRP each have different mechanisms and effects. Would it be possible for the authors to perform a subgroup analysis or additional analysis to address these concerns?” – We agree that the classification of groups in this NMA limits the ability to effectively assess targeted treatment effects of individual interventions that may have different mechanisms and effects if considered separately. However, given the small number of studies, a sensitivity analysis would not be statistically viable. For example, for injections as grouped right now, we have 2-3 studies directly comparing ROM, Pain, SPADI and DASH. If we re-classify, we will either not be able to perform an NMA or the robustness and reliability of results can be compromised. Moreover, included studies frequently grouped electrotherapy, thermotherapy, and dry needling as one group (i.e., Physical Therapy group), so we decided to treat these interventions as one group for consistency. For injections, we grouped all types of injections in one group considering that all injections are meant to decrease pain indifferently of the type of injection applied. As the research in this area increases and more studies are published, subset analyses will be possible. We add this limitation in the limitation session of the revised manuscript (Lines 374-379). 

5. “Lastly, could you please provide the inconsistency results of the NMA?” – Results were added to the revised manuscript (lines 266-267, 276-277, 297-300). Full results were also incorporated to the revised supporting material (Pages 44-50).

Reviewer #2: No queries.

Reviewer #3: 

1. “Consider describing the objective of the research clearly, because there are two different objectives throughout the publication, this can be reviewed in line 5 and line 74.” – Changes made as requested. Please see revised manuscript (Lines 75-77).

2. “Consider mentioning the definition of: "usual medical care" in the introduction and not only in the discussion on line 307.” - Changes made as requested. Please see revised manuscript (Lines 55-56).

3. “Regarding the conclusion, consider answering the objective of the research by modifying the approach of the objectives mentioned above.” - Changes made as requested. Please see revised manuscript (Lines 414 and 416).

We believe that these revisions have significantly strengthened the manuscript by clarifying the manuscript objectives, methods, results and limitations. We appreciate the time and effort you have dedicated to the peer review process, which has undoubtedly improved the quality of our work. If you have any additional comments or concerns, please do not hesitate to contact us.

Thank you once again for your invaluable feedback and for your commitment to advancing the quality of research in our field. We look forward to your positive response and the possibility of seeing our work published in PLOS ONE.

Sincerely, 

Anelise Silveira, PT, MSc, PhD Candidate

REFERENCES

1. Hickey D, Solvig V, Cavalheri V, Harrold M, McKenna L. Scapular dyskinesis increases the risk of future shoulder pain by 43% in asymptomatic athletes: a systematic review and meta-analysis. Br J Sports Med. 2018;52(2):102-10.

2. Salamh PAH, W.J.; Boles, T.; Holmes, D.; McMillan, A.; Wagner, A.; Kolber, M.J. Is it Time to Normalize Scapular Dyskinesis? The Incidence of Scapular Dyskinesis in Those With and Without Symptoms: a Systematic Review of the Literature

. International Journal of Sports Physical Therapy. 2023;18(3):558-76.

---

## [Editor Report · Decision Letter 1]

25 Oct 2023

Shoulder specific exercise therapy is effective in reducing chronic shoulder pain: A network meta-analysis

PONE-D-23-11628R1

Dear Dr. Silveira,

We’re pleased to inform you that your manuscript has been judged scientifically suitable for publication and will be formally accepted for publication once it meets all outstanding technical requirements.

Kind regards,

Juan Ignacio Guerrero-Henriquez, MSc

Academic Editor

PLOS ONE
---

## [Editor Report · Acceptance letter]

27 Feb 2024

PONE-D-23-11628R1 

PLOS ONE

Dear Dr. Silveira, 

I'm pleased to inform you that your manuscript has been deemed suitable for publication in PLOS ONE. Congratulations! Your manuscript is now being handed over to our production team.

Kind regards, 

on behalf of

Mr. Juan Ignacio Guerrero-Henriquez 

Academic Editor

PLOS ONE